# Gene Networks and Pathways Involved in LPS-Induced Proliferative Response of Bovine Endometrial Epithelial Cells

**DOI:** 10.3390/genes13122342

**Published:** 2022-12-12

**Authors:** Mojtaba Najafi, Yongzhi Guo, Göran Andersson, Patrice Humblot, Erik Bongcam-Rudloff

**Affiliations:** 1Department of Animal Sciences, Gorgan Agricultural Sciences and Natural Resources University, Golestan 4913815739, Iran; 2Department of Animal Breeding and Genetics, Swedish University of Agricultural Sciences (SLU), P.O. Box 7070, SE-750 07 Uppsala, Sweden; 3Department of Clinical Sciences, Swedish University of Agricultural Sciences (SLU), P.O. Box 7070, SE-750 07 Uppsala, Sweden

**Keywords:** cell proliferation, lipopolysaccharide, bovine endometrium epithelial cells, RNASeq

## Abstract

Lipopolysaccharide (LPS) is a component of the outer membrane of Gram-negative bacteria involved in the pathogenic processes leading to mastitis and metritis in animals such as dairy cattle. LPS causes cell proliferation associated with endometrium inflammation. Former *in vitro* studies have demonstrated that LPS induces an intense stimulation of the proliferation of a pure population of bovine endometrial epithelial cells. In a follow-up transcriptomic study based on RNA-sequencing data obtained after 24 h exposure of primary bovine endometrial epithelial cells to 0, 2, and 8 μg/mL LPS, 752 and 727 differentially expressed genes (DEGs) were detected between the controls and LPS-treated samples that encode proteins known to be associated with either proliferation or apoptosis, respectively. The present bioinformatic analysis was performed to decipher the gene networks involved to obtain a deeper understanding of the mechanisms underlying the proliferative and apoptosis processes. Our findings have revealed 116 putative transcription factors (TFs) and the most significant number of interactions between these TFs and DEGs belong to *NFKβ1*, *TP53*, *STAT1*, and *HIF1A*. Moreover, our results provide novel insights into the early signaling and metabolic pathways in bovine endometrial epithelial cells associated with the innate immune response and cell proliferation to *Escherichia coli*-LPS infection. The results further indicated that LPS challenge elicited a strong transcriptomic response, leading to potent activation of pro-inflammatory pathways that are associated with a marked endometrial cancer, Toll-like receptor, NFKβ, AKT, apoptosis, and MAPK signaling pathways. This effect may provide a mechanistic explanation for the relationship between LPS and cell proliferation.

## 1. Introduction

Bacterial infections in cattle lead to diseases such as mastitis (inflammation of the mammary gland) and endometritis (inflammation of the endometrium), which are ranked as top diseases and major economic burdens in the dairy cattle industry [2]. Based on financial calculations, the cost of these infections has been estimated at USD 650 million in the USA and EUR 1.4 billion in the European Union dairy industry per annum [3]. The cost of mastitis has varied largely in previous studies. Per-cow base mastitis cost varied from USD 2.27 in Scotland to USD 35.1 in Ohio [4]. Previous studies have shown that uterine infections lead to metritis in 40% of animals within a week of parturition, persisting as endometritis in 20% of animals for several weeks [5]. Bacteria that infect mammals are identified by the infected host’s pattern recognition receptors on the innate immune system cells based on recognizing their pathogen-associated molecular patterns [6]. The Gram-negative bacteria consist of a large group of which several species are considered as primary pathogens causing severe infections [7]. Lipopolysaccharide (LPS), a bacterial endotoxin, is a major component of the outer membrane of Gram-negative bacteria such as *E. coli* [8]. Its accumulation in follicular fluid in the animal is evidence of microbial invasion and a considerable pro-inflammatory response that actives different pathways and ultimately leads to the release of a large number of pro-inflammatory cytokines and chemokines [8].

Numerous studies have shown that pro-inflammatory cytokines contribute to the development of chronic mastitis, reduce circulating and intrafollicular estradiol concentrations, extend luteal phases, disrupt ovarian cyclic activity, and change nutritional metabolism, cell proliferation, and apoptosis. These inflammatory responses negatively affect fertility and have a further negative effect on milk yield, fat content, and milk energy efficiency [9,10,11,12]. The epithelial cells are the first cellular barrier to infection and play a key role in alerting the immune system to infection and respond by secreting pro-inflammatory cytokines and chemokines [13]. Increased cell proliferation is part of the inflammatory process and has been reported in human epithelial and immune cells [14,15,16] and in bovine endometrial epithelial cells [17]. There are several reports concerning the effect of LPS on different tissues in various animal species [17,18,19,20,21,22,23,24]. Most of these studies performed in different species have shown inconsistent actions of LPS, including either stimulation or inhibition of proliferation. However, few studies have described the possible effect of factors related to the host on cell proliferation and their role as a source of variation in this response. In addition, according to our knowledge, the specific mechanisms by which LPS affects proliferation have not been well-documented for bovine endometrial epithelial cells. Moreover, previous studies have proposed that the presence of cell proliferation in tissues or organs with low mitotic activity could be a significant factor for cancer development in human [25].

Taking the above information into account, cell proliferation plays an important and even critical role in many steps in cancer development, such as initiation, promotion or selection, and progression during further invasion in several organs and tissues. Previous studies have noted the stimulation of proliferation despite 50% of genes related to the proliferation process being either under- or overexpressed regulated genes in bovine endometrial epithelial cells (bEECs) after being exposed to different doses of LPS [1,17,26]. The present study aimed to decipher the corresponding mechanisms considering these observations and assess the gene network and inter-cellular signaling pathways leading to cell proliferation to obtain a deeper understanding of the mechanisms underlying the proliferative and apoptosis processes.

## 2. Materials and Methods

### 2.1. Experiment Design

The endometrial epithelial cells were purified from uterine horns of three Swedish Red Breed cows according to procedures previously described [1]. Briefly, tissue pieces were incubated with collagenase IV (C5138, Sigma, Saint Louis, MO, USA) and hyaluronidase (250 U/mL) (H3506, Sigma) in PBS containing 2% bovine serum albumin (BSA) (Sigma). Epithelial cells were collected from the filtration through a 250 μm gauze (to remove mucus and undigested tissue) and a 40 μm nylon sieve (which allowed the fibroblasts and blood cells to pass through while epithelial cells were retained). The purity of bEEC culture was estimated by morphological observation following three to five passages. The purity of the epithelial cell culture was then checked by flow-cytometry (Appendix A). From passage two and thereafter, more than 98% of cells expressed cytokeratin, confirming the very high purity of the cell culture system (Appendix A). bEECs were challenged on passage five, with 0, 2, and 8 μg/mL LPS from *E. coli* (O111:B4; Sigma). Therefore, the bEECs were classified into four groups, including Control 0 h, Control 24 h, LPS-2 µg/mL 24 h, and LPS-8 µg/mL 24 h as previously described [1]. The change in the cellular biology and physiology of bEECs following LPS were shown in our previous work [17].

Then, RNA was prepared from these samples to be used for RNASeq analysis. Sequencing libraries from 12 samples (for each cow, on Time 0, and with 0, 2, and 8 μg/mL LPS at 24 h) (Appendix A) were prepared by a sequencing platform (Science for Life Laboratory, Uppsala University; https://www.scilifelab.se/ accessed on 12 March 2016) using the Illumina HiSeq2500 system (Illumina Inc. San Diego, CA, USA). The RNASeq raw data were uploaded to the European Nucleotide Archive (ENA) with the accession number PRJEB34011. In a follow-up transcriptomic study from RNA sequencing [1] among nearly 26,000 genes annotated in the bovine genome, by taking comparisons between control 24 h vs. 2 µg/mL of LPS, 2035 differentially expressed genes (DEGs) were identified based on the Benjamini–Hochberg (BH) correction and adjusted *p*-values < 0.05 and log 2-fold change ≥ 0 for over- or underexpression as the criteria for defining differentially expressed genes. In the following, these DEGs were evaluated based on different fold-change for detecting the most effective DEGs.

### 2.2. Bioinformatics Analysis (Classification of DEGs Involved in Cell Proliferation)

To determine the possible cellular function involved in the LPS-induced proliferation of bEECs, a gene classification analysis of DEGs was performed using two databases including Mitocheck (https://www.mitocheck.org/ accessed on 30 May 2018) and GeneCards (https://www.genecards.org/ accessed on 28 May 2017). The obtained gene sets (common genes between 2035 DEGs and databases) associated with biological pathways were considered for literature analysis.

### 2.3. Transcription Factor Analysis

The OPOSSUM (https://bio.tools/opossum/ accessed on 12 July 2018) and Genomatix databases (https://www.genomatix.de/ accessed on 13 July 2017) for identification of transcription factor (TFs) target genes were applied. The genes from the datasets associated with canonical pathways in these databases were considered for the identified TF-predicted target genes analysis. After uploading the datasets, TFs identifiers were mapped to corresponding gene objects.

### 2.4. Functional Annotation and Pathways Analysis

Gene ontology and KEGG pathway enrichment were determined using DAVID 6.7 using all Ensembl genes as background and also the Genebrowser platform (http://bioinformatics.ua.pt/genebrowser2/ accessed on 20 July 2018). For functional annotation of the significant genes the Database for Annotation, Visualization and Integrated Discovery (DAVID; http://david.abcc.ncifcrf.gov/ accessed on 22 July 2018) was applied based on a modified Fisher’s exact *p*-value to demonstrate gene ontology (GO) or molecular pathway enrichment. *p*-values less than 0.05 were considered to be significantly enriched in the annotation category.

## 3. Results

### 3.1. RNASeq Analysis

Our previous RNASeq study generated approximately 27 million reads (paired-end 125 bp reads) per sample following quality control [1]. Ninety-seven percent of these reads mapped to the UMD v3.1.1 bovine reference genome (Appendix A). There are a few DEGs (with low fold change) between the Control 0h and Control 24 h groups (Figure 1). Compared to Control 24 h, exposure to the LPS revealed the differential gene expression of 2035 transcripts at 2 µg/mL LPS-induced (248 increased and 40 decreased in expression > 1 log2_fold-change) and 2073 transcripts at 8 µg/mL LPS-induced. In addition, no differential gene expression was identified between the two mentioned LPS-induced groups (*p* > 0.05), indicating that in almost all cases, gene expression levels for these genes show similar variation in response to LPS for both treatment groups. This was further confirmed by the correlation coefficient between fold changes observed with 2 and 8 μg/mL LPS is *r*^2^ = 0.99. Therefore, the 2035 DEGs between control 24 h and LPS-2 μg/mL were further analyzed.

### 3.2. Results of the Comparison between Control 24 h and LPS-2 µg/mL 24 h

These 2035 DEGs between control 24 h and LPS-2 µg/mL 24 h were searched and classified for in the proliferation databases Mitocheck and GeneCards. Based on this database search, we obtained a dataset consisting of 752 genes classified to be involved in cell proliferation. Out of these genes, 400 were overexpressed, and 352 genes were underexpressed. Table 1 illustrates the intensity of differential expression of these genes according to over- or under-expression. The proportion of genes with differential expression > 1 log2_fold-change was higher for overexpressed genes than for under-expressed genes (138/400 vs. 13/352; *p* < 0.00001).

Among the genes displaying the most dramatic changes in levels due to LPS exposure were those involved in the inflammatory response and immune response: cytokines or chemokines, interferon-related genes, interleukin-related genes, TNF-related genes, growth factors, and prostaglandin-related genes. Additionally, differential gene expression was observed for some classical markers of cell proliferation, including *IL-8*, *IL-6*, *CCL2*, *TNF*, *CXCL6*, *P53*, *Ki-67*, *BCL-2*, *CCL5*, *AP-1*, *Cyclin D3*, and *PCNA*.

The top 12 overexpressed genes (≥3-fold) after the LPS challenge associated with cell proliferation are listed in Table 2. All genes encode proteins that play established roles in immune response and cell proliferation during infection including genes encoding cytokines (*IL-1A*, *CXCL8*, *CXCL6*, *TNF*, and *BCL2A1*) and anti-microbial defense (*C3*) *SAA3* (an acute-phase protein), as well as those involved in the metabolism pathway (*SLC5A5*).

The gene ontology analysis showed that the majority of under-expressed genes encode proteins that are mainly involved in cell structure and cell adhesion while the overexpressed genes encode proteins that are involved in immune response and cell proliferation or apoptosis.

The GO molecular function analysis uncovered an increase in genes encoding proteins with known critical functions in regulating the cell proliferation such as *BCL2*, *BCL3*, *LIF*, *CD14*, *CyclinD3*, and *RAS*.

### 3.3. Classification of DEGs Based on Their Functions in Cell Cycle

Based on the importance of cell proliferation and apoptosis, more analyses were conducted using the GeneCards and Mitocheck databases. In these comparative analyses, we observed the common genes between our DEGs (2035) and the genes which exist in both databases according to their functions (Table 3 and Table 4). The results show the most significant number of genes involved in cell differentiation and cell growth (Table 3).

Further analysis was performed to classify the gene sets based on their influence on phenotypes using the Mitocheck database (Table 4). In this database, genes are submitted based on phenotype data. The results show the most significant number of genes involved in mild inhibition of secretion, such as A-kinase anchoring proteins (*AKAP*), AKT serine/threonine kinase, and protein kinase C γ (*PRKCG*) (Table 4).

### 3.4. Differential Expression of Multiple Transcription Factor Families

Two databases were used to find genes encoding TFs among the 752 DEGs involved in proliferation. Based on the results from OPOSSUM database searches, among these 752 genes, 161 genes were recognized as encoding TFs. Moreover, these genes were classified by the TFs encyclopedia from the OPOSSUM database into seven different classes (Figure 2). Most of the TFs identified here belonged to the zinc-coordinating class (Figure 2).

Additionally, genes encoding multiple TFs were identified among the differentially expressed genes associated with cell proliferation. These transcription factors, including *TP53*, *STAT*, *E2F*, *MYC*, *AP-1*, *NFKB*, and *MYB*, are essential in either proliferation or apoptosis processes. In addition, RNASeq reads revealed that *JunB*, *JunD*, *Foxp4*, and *Ets2* were particularly overexpressed in the epithelial cells. We found that the expression changes in the majority of the observed cytokines and chemokines were directly regulated by the identified TFs, including *STAT1*, *NFKB*, and *TP53*.

In addition, we evaluated the interaction between TFs and DEGs using the Genomatix database. Here, we detected 52 TFs and their interactions with genes (Table 5). Among these TFs, a large number of interactions belonged to NFKB1, TP53, STAT1, and HIF1A. Some of these TFs have predicted or validated binding sites and others have interaction evidence.

### 3.5. Biological Pathway Analysis

Within this set of DEGs, we identified 50 significantly enriched canonical pathways using the Genebrowser platform (Available online: https://bioinformatics.ua.pt/genebrowser/ accessed on 30 July 2018). The majority of which were signaling pathways (Appendix A). It was striking that signaling pathways related to immune or inflammatory functions were the most affected in our study. Interestingly, among the signaling pathways, our data revealed induction of specific pathways related to different carcinoma such as endometrial, lung, renal cell, pancreatic, colorectal, and prostate cancer was coupled with activation of certain pathways, such as ERK/MAPK and PI3K/AKT signaling related to cell proliferation. More intriguing is the finding that among all of the biological pathways, endometrial cancer is more related to cell proliferation in bEECs following LPS challenge (Figure 3). Additionally, we depicted a pathway based on our results that shows the interaction between over- and under-expressed genes after LPS challenge in bEECs (Figure 4).

## 4. Discussion

Previous studies have shown that LPS can affect cell proliferation through different pathways in several different tissues [18,27,28,29]. Numerous studies have revealed that LPS stimulates inflammation responses [26,30]. At the same time, complex negative feedback regulatory networks are acting to obviate inflammation responses [31]. The present study, based on the change in gene expression profile, shows that bEECs have been activated by LPS challenge, which is fully consistent with previous studies [5,17,26,32]. So far, several intercellular signaling pathways have been identified [33,34], but further research is needed to fully understand their mechanisms. In the present study, we evaluated the change in gene expression and the pathways involved in cell proliferation and apoptosis in bEECs following *E. coli* LPS challenge. There are numerous reports about the effect of LPS on cell proliferation in human tissues and cell types, including immune cells [35], nasal mucosa [36], endothelial cells [23], hepatic stellate cells [37], neural cells [22], fibroblast cell line [18,21], tumor cells [38], bronchial epithelial cells or cell lines [14,39], gastric epithelial cells [19], intestinal cells and cell lines [27], small epithelial intestinal cells [40], colon epithelial cells [32], cholangiocytes [41,42,43], corneal epithelial cells [20], pulmonary microvascular endothelial cells [28], and alveolar type II cells [44].

Despite the abovementioned studies that have shown a significant association between LPS challenge and an increase in cell proliferation, some reports have indicated the inhibition role of LPS in cell proliferation, such as a decrease in viability for human epithelial cells and gingival fibroblasts [45], a reduction in the proliferation of mammary epithelial cells [29] and also the growth inhibition and increased apoptosis of peritoneal mesothelium cells [46]. Additionally, a previous study reported no significant effect of LPS on cell proliferation in pig intestinal epithelial cells [47]. In contrast, a comparative analysis of LPS effects on cell proliferation in different types of epithelial cells shows that LPS has an inhibitory role in cell proliferation and growth, including inhibition of proliferation of rat tracheal epithelial cells [48], and rat kidney epithelial cells [49,50], inhibition of taste progenitor cell proliferation (mouse) [30], inhibition of proliferation in a rat tongue epithelial cell line [51], and even differential expression of the genes associated with cell growth and proliferation in tammars, macropod marsupials (*Macropus eugenii*) [52].

LPS was reported to inhibit the proliferation of post-primary bovine mammary epithelial cells but not in bovine mammary cell lines [53]. However, in another study, LPS increased the number of bovine mammary epithelial cells [54]. Ultimately, these findings have shown that there is no consistent response of cells following LPS challenge. The effect of LPS on cell proliferation will differ depending on species, tissues, study model (*in vivo* or *in vitro*), and, importantly, also the time and dose of administered LPS. The effect of different doses of LPS on bovine endometrium epithelial cells has previously been carefully evaluated by our group [17]. A significant increase in the number of cells for dosages 2, 4, 6, 8, and 12 μg/mL compared to the control group was shown. With higher dosages, the number of live cells did not increase, but the number of dead cells increased. Decreasing cell proliferation in high LPS doses may be due to the induction of apoptosis.

Recently, in a genome-wide DNA methylation study on the bEECs we noted a significant overrepresentation of differentially methylated regulatory promoters of genes encoding proteins of biological and molecular functions related to cell proliferation, and apoptotic process [26] which is consistent with transcriptome data previously obtained by RNASeq on the same cell samples [1].

Moreover, our results confirmed that the bEECs act as a mechanical barrier, and also contribute to innate immunity by reacting to LPS with the secretion of pro-inflammatory chemokines/cytokines and the induction of proliferation. This wide array of cytokines and interferon-related genes demonstrates the ability of the epithelial cells to recognize and react to LPS. In our findings, differential gene expression was observed in some known markers for cell proliferation, including *IL-8*, *IL-6*, *CCL2*, *TNF*, *CXCL6*, *Complement C3*, *P53*, *Ki-67*, *BCL-2*, *CCL5*, *AP-1*, *Cyclin D3*, and *PCNA*. Recently, a study demonstrated that *CXCL6* (Chemokine (C-X-C Motif) ligand 6) knockdown significantly inhibits human brain microvascular endothelial cells proliferation by modulating Sirt3 expression through inactivation of AKT/FOXO3a [55]. In addition, complement C3 expression was associated with mouse T cell proliferation and IL-17A expression, which was mediated via ERK and STAT3 signaling pathways [56].

In the present study, the GO analysis showed that under-expressed DEGs such as *PTHLH* and *ECM2* encode proteins that are mainly involved in cell structure and cell adhesion. Most extracellular matrix protein 2 (ECM2) glycoproteins promote cell adhesion and cell survival [57]. In addition, PTHLH plays a central role in the physiological regulation of bone formation [58]. The GO molecular function analysis uncovered an increase in encoding genes such as *BCL2*, *BCL3*, *LIF*, *CD14*, *CyclinD3*, and *RAS*, with the latter being necessary in regulating cell proliferation. In the GO term, LIF protein is considered as “negative regulation of cell proliferation”. Functionally, suppressed LIF increases the proliferation rate and migration ability of cells [59].

Our detected TFs were classified into three groups including TF classes with 1–10 TFs such as β-sheet, lg-fold, other α-Helix classes; TF classes with 10–20 TFs such as winged helix–turn–helix, helix–turn-helix, zipper-type; and TF classes with more than 20 TFs such as zinc-coordinating. Zinc finger proteins, the most abundant and structurally diverse groups of protein in nature, act as TFs and function in different cellular processes, including cell proliferation [60,61,62]. B-cell lymphoma/leukemia 11B (BCL11B) is one member of the large family of zinc finger proteins [61]. Its inhibition results in growth retardation and apoptosis in T-cells [63]. The most relevant TFs include c-Jun, ATF-2, PPAR-γ1 and 2, NFKB1 and 2, and STAT3, which all are involved in regulating cell proliferation processes. A previous study has shown that the induction of AP-1 causes cell proliferation and organ growth [64]. A recent study also noted that the cell proliferation was regulated by direct targeting c-Jun mRNA in gastric carcinoma [65]. Therefore, it is clear that among all the TFs identified, only some of them will contribute to proliferation or an apoptosis signature. This is also why in-depth analyses in different species and tissues are critical to define expression specificities before further biological investigations are performed. Our study has revealed 116 putative TFs being DEGs in response to LPS and the most significant number of interactions between TFs and DEGs belonged to NFKB1, TP53, STAT1, and HIF1A. Taken together, these findings specify that multiple TF families might be involved in regulating the activation of bEECs in response to LPS, which helps us to deeper understand cell proliferation/apoptosis mechanisms in bEECs following exposure to LPS.

## 5. Conclusions

A disturbance in the bovine endometrium-specific expression of cytokines, chemokines, growth factors, steroid hormones and their nuclear receptors, gonadotropins and their receptors, and heat shock proteins in response to LPS probably triggers the multiple factors in the epithelial cells that are responsible for cell proliferation or apoptosis.

Moreover, our results provide novel insights into the early signaling and metabolic pathways in bovine endometrial epithelial cells associated with the innate immune response and cell proliferation to *E. coli*-LPS infection. The results further indicated that LPS challenge elicited a strong transcriptomic response, leading to potent activation of pro-inflammatory pathways that were associated with a marked endometrial cancer, Toll-like receptor, NFKβ, AKT, apoptosis, and MAPK signaling pathways. This effect may provide a mechanistic explanation for the relationship between LPS and cell proliferation.

## Figures and Tables

**Figure 1 genes-13-02342-f001:**
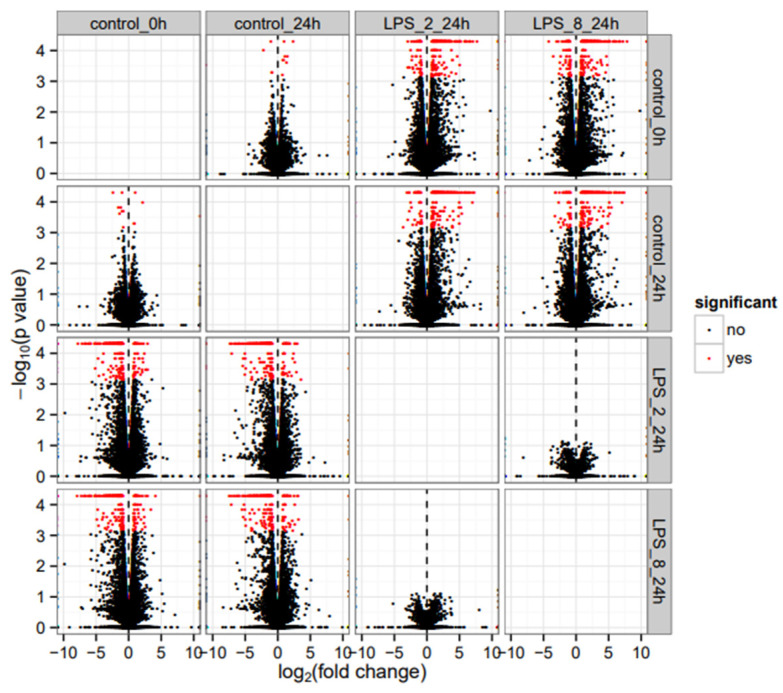
Comparison between different groups following LPS challenge including Control 0 h, Control 24 h, LPS-2 µg/mL 24 h, and LPS-8 µg/mL 24 h. Dots in red show DEGs (adjusted *p* value < 0.05 from Benjamini-Hochberg correction). Significant means positive log2 fold changes correspond to increased expression, whereas negative values correspond to decreased expression.

**Figure 2 genes-13-02342-f002:**
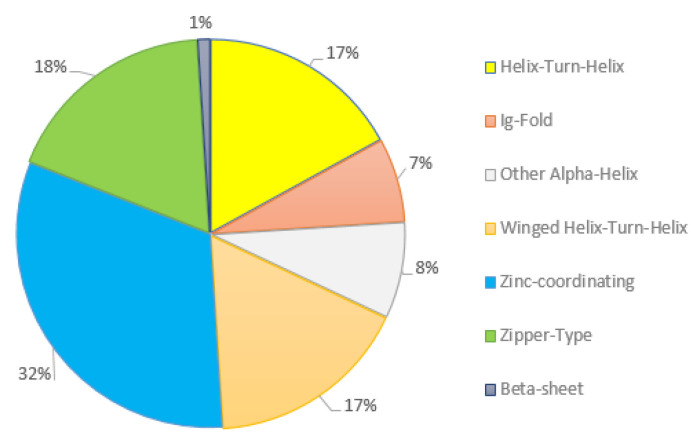
Classes of 116 putative expressed TFs in bovine endometrial epithelial cells.

**Figure 3 genes-13-02342-f003:**
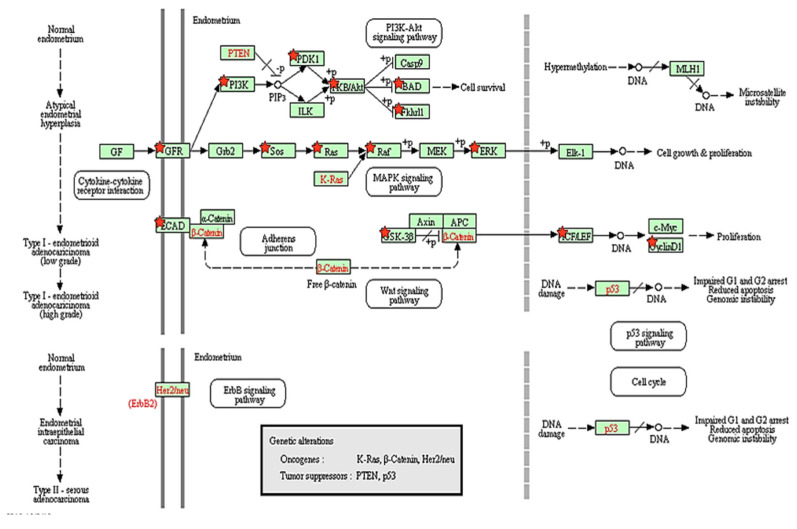
Endometrial cancer pathway obtained from detected DEGs following LPS challenge. Red asterisks (
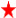
) show DEGs in our RNASeq reads. The red font in the figure shows tumor suppressor genes.

**Figure 4 genes-13-02342-f004:**
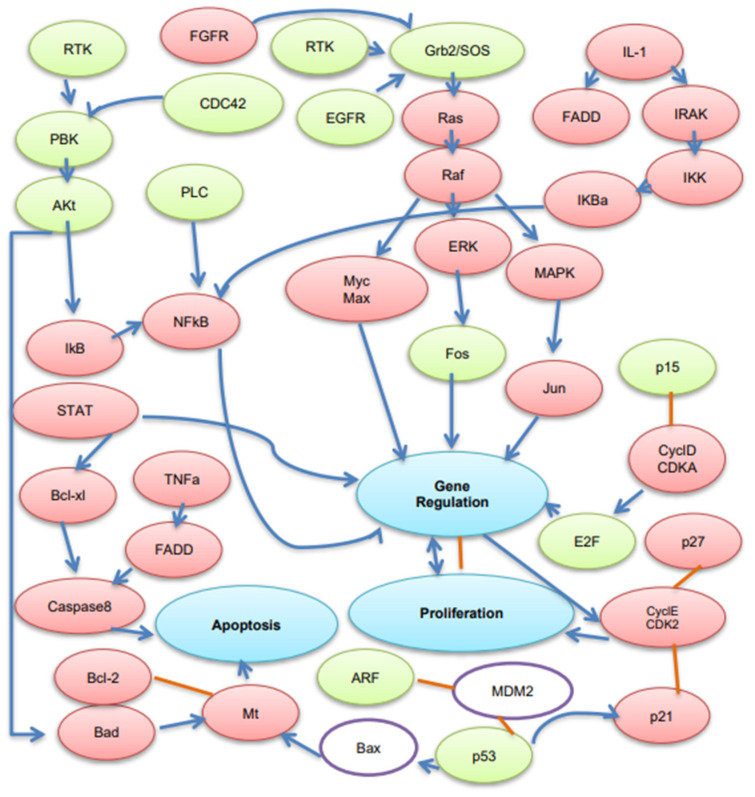
Biological pathways related to cell proliferation based on our results. Red denotes upregulated genes and green denotes downregulated genes. *MDM2* and *Bax* (purple circles) were listed in published studies but were not among our detected DEGs. An arrow pointing (blue lines) signifies activation (e.g., binding, phosphorylation); orange lines signify direct protein interactions.

**Table 1 genes-13-02342-t001:** Distribution of detected DEGs involved in cell proliferation based on different log2-fold-change.

Function	State	Fold-Change	Number	Ratio (Up/Down)
Proliferation (752)	Over-expressed	>2 log2_fold-change	28	1.14
		>1 log2_fold-change	110	
		>0 log2_fold-change	400	
	Under-expressed	>1 log2_fold-change	13	
		>0 log2_fold-change	352	

**Table 2 genes-13-02342-t002:** List of top 12 overexpressed genes (≥3-fold change) in detected DEGs related to cell proliferation and apoptosis.

Gene Symbol	Gene Name	Known Function	Fold Change
*CXCL6*	Granulocyte chemotactic protein 2	Cytokine and chemokine activity, strong antibacterial activity	6.35
*C3*	Complement component 3	Activation of complement system to form mature proteins, modulates inflammation and possesses antimicrobial activity, activation of the PLC, MAPK, and AKT signaling pathways	4.55
*BCL2A1*	BCL2-related protein A1	Anti- and pro-apoptotic regulators, lymphocyte activation as well as cell survival	4.21
*SLC5A5*	Solute carrier family 5	Thyroid hormone synthesis and metabolism pathway. Increased viability	3.79
*LGALS9*	Lectin, galactoside-binding, soluble, 9	Enhancing cell migration, inhibits angiogenesis, activates ERK1/2 phosphorylation inducing cytokines (IL-6, IL-8, and IL-12) and chemokines (CCL2)	3.49
*CXCL8*	Interleukin 8	Chemotaxis; neutrophil activation; G-protein coupled receptor protein signaling pathway; angiogenesis	3.35
*CTSC*	Cathepsin C	Activation of many serine proteinases in cells of the immune system, protein binding.	3.32
*CX3CL1*	Chemokine (C-X3-C motif) ligand 1	Receptor binding and chemokine activity; regulating leukocyte adhesion and migration processes	3.32
*CXCL3*	Chemokine (C-X-C motif) ligand 3	Inflammation, chemokine activity, and CXCR chemokine receptor binding	3.23
*CCL5*	Chemokine (C-C motif) ligand 5	Immune-regulatory and inflammatory processes, activation of the PI3, Akt, and MAP kinases	3.16
*TNF*	Tumor necrosis factor	Acute phase response, pro-inflammatory immune response, regulation of cytokine secretion, insulin signaling, and glucose metabolism	3.06
*IL1A*	Interleukin 1 α	Immune responses, inflammatory processes, and cell proliferation. Stimulates the release of prostaglandin and collagenase.	2.86

**Table 3 genes-13-02342-t003:** Detected functions of the common genes involved in cell cycles using the GeneCards database.

Functions	Database	Common Genes
Tumor suppressor genes	3472	488
Segregation problems	829	111
Metaphase delay	217	33
Cell death	5722	731
Metaphase alignment problem	25	5
Condensation followed by decondensation	43	9
Binuclear	584	66
Dynamic change	3679	524
Mitotic delay	1456	212
Migration (speed)	2877	317
Migration (distance)	300	46
Inhibition of secretion	4027	583
Enhanced secretion	3135	479
Failure in decondensation	21	1
Chemokine	1258	201
Cytokines	3324	513
Stay close together	95	20
Strange nuclear shape	98	8
Transcription Factor	10943	1141
DNA replication	4260	535
Spindle mitotic	1329	183
Cell division	1527	228
Cell growth	10029	1147
DNA damage	5377	693
Mitochondrial respiration	429	63
Electron acceptors	203	36
Endometrial cancer	1503	288
Proteasome phosphorylation	2456	374
Telomerase erosion	151	36
Chromosome duplication	3058	274
Centrosome duplication	408	56
Cell cycle	6332	751
Endothelial cell	4650	718
Cell differentiation	10738	1223
VEGF signaling	2892	492
Steroid hormone receptors	1558	238
Interphase	725	74
Prophase	378	39
Metaphase	686	82
Anaphase	754	85

**Table 4 genes-13-02342-t004:** Classification of DEGs based on their phenotypic effects using the Mitocheck database.

Phenotypes	Database	DEGs
Enhanced secretion	223	19
Inhibition of secretion	783	75
Mild inhibition of secretion	2306	225
Strong inhibition of secretion	1524	146
Increased proliferation	96	10
Migration (distance)	144	15
Migration (speed)	277	20
Grape	153	14
Mitotic delay	443	49
Dynamic changes	741	71
Large	316	25
Polylobed	472	44
Binuclear	456	53
Condensation followed by decondensation	10	1
Failure in decondensation	8	0
Metaphase alignment problems	422	53
Cell death	782	80
Metaphase delay	275	34
Segregation problems	494	65
Strange nuclear shape	583	70
Nuclei stay close together	364	47
Altered gm130 morphology	99	16
Altered COPI morphology	108	12
Altered COPII morphology	59	6
Retention of sh4(haspb)-gfp	302	33
Retention of sh4(yes)-mcherry	126	11
Reduction in ir-induced 53bp1	3	0
Accumulation of gfp-rnf168 on	34	3

**Table 5 genes-13-02342-t005:** List of TFs and their interaction with genes using the Genomatics database.

Transcription Factors	Interaction	Transcription Factors	Interaction
TRIM24 (transcription cofactor)	13	XBP1 (V$CREB)	60
LTF (V$LTFM)	40	PML (transcription cofactor)	82
BATF2 (V$AP1F)	7	HOXC4 (V$HOXF, V$HOXC)	7
PIAS1 (transcription cofactor)	29	OSR1 (V$OSRF)	3
EYA3 (transcription cofactor)	2	FOX1 (V$FKHD)	36
GZF1 (V$GZF1)	0	JARID2 (V$ARID)	5
CCNT1 (transcription cofactor)	14	CNOT8 (transcription cofactor)	5
ZNF217 (V$ZF03)	21	PPARGC1B (transcription cofactor)	22
CALR (transcription cofactor)	66	CREM (V$CREB)	26
BHLHE41 (V$HESF)	12	NFKB1 (V$NFKB)	279
ETV3 (V$ETSF)	3	RELB (V$NFKB)	71
MTA2 (transcription cofactor)	18	NFE2L1 (V$TCFF, AP1R)	6
CIITA (transcription cofactor)	50	CDCA7	0
HIF1A (V$HIFF)	155	FOSL1 (V$AP1F)	66
MED17 (transcription cofactor, mediator)	3	FOXO1 (V$FKHD)	84
IRF1 (V$IRFF)	111	NRIP1 (transcription cofactor)	28
KDM2A (transcription cofactor, demethylase)	4	PRDM1 (V$PRDF)	39
BCL3 (transcription cofactor)	55	PCBD1 (transcription cofactor)	4
MEF2D (V$MEF2)	13	TB53 (V$P53F)	254
EHF (V$ETSF)	17	NFKB2 (V$NFKB)	59
STAT1 (V$IRFF, V$STAT)	157	SMAD3 (V$SMAD)	98
SRXN1 (V$SNAI)	18	SIX5 (V$MEF3)	2
ELL3 (transcription elongation cofactor)	1	MYCL (V$EBOX)	13
ARRB1 (transcription cofactor)	37	JUNB (V$AP1F)	87
PBX4 (V$PBXC, V$HOXC)	0	NFE2L3 (V$AP1R)	9
AHR (V$AHRR)	84	LITAF	19

## Data Availability

Not applicable.

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
