# Peer review of "Gene Networks and Pathways Involved in LPS-Induced Proliferative Response of Bovine Endometrial Epithelial Cells"

_genes, 2022, doi:10.3390/genes13122342_

Round 1
Reviewer 1 Report
The submitted manuscript titled "Gene networks and pathways involved in LPS-induced proliferative response of bovine endometrial epithelial cells" by Najafi et al. first attracted my attention. The authors evaluated the transcriptomic changes in bovine endometrial epithelial cells after 24 h of a challenge using two concentrations of LPS (2 and 8 μg/mL). In addition, the differentially expressed genes between Control 24 h and 2 μg/mL LPS groups were further analyzed.
In general, such research to better understand cellular processes in gram-negative uterine infections is very important to improve fertility rates.
However, the presented study shows essential weaknesses in describing the experiments and presenting the results. Furthermore, the manuscript must be improved concerning coherence and cohesion. The manuscript lacks good presentation and seems that has not been carefully revised by the authors.
In addition, the manuscript needs standard English revision to correct some grammatical mistakes. Please, consider assistance from a native English speaker concerning grammar and some syntax errors.
Therefore, the manuscript in its present form is not suitable for publication.
Specific points:
Abstract:
1. P1/L18: The abbreviation “bEECs” is no longer used in the abstract section. Please, consider removing it.
2. P1/L26: “Escherichia coli” instead of “E. coli”
Introduction:
1. P1/L34: The sentence “Bacterial infections of the uterus lead to diseases such as mastitis and endometritis, (…)” is incorrect. Mastitis is an inflammation of the mammary gland mainly caused by an infection, but it is not a bacterial infection of the uterus. Please, rephrase the sentence and review the Introduction section accordingly.
2. P1/L35: Why is the first reference listed as number 2?
3. P1/L43: The abbreviation “PAMPs” has no longer been used in the manuscript. Consider removing it.
4. P1/L41-43: Please, provide references for the statement “Bacteria that infect mammals (…) pathogen-associated molecular patterns (PAMPs).”
5. P1-2/L45-47: Please, provide references for the statement “Lipopolysaccharide (LPS), a (…) such as Escherichia Coli.”
6. P2/L46-47: Use italics in the species name. Use lowercase letters for “Coli”.
7. P2/L55-57: Please, provide references for the statement “The epithelial cells are the first (…) and secreting pro-inflammatory cytokines and chemokines.”
8. P2/L67: “tissues” instead of “tissue”
9. P2/L67: “organs” instead of “organ”
Material and Methods:
1. P2/L80-81: How did you obtain the endometrial epithelial cells? Did you obtain the cells in vivo? From slaughterhouse by-products? Did you purchase or purify the LPS used? There are many unanswered questions regarding the experimental design of the study. Please, provide more information about the study's experimental design and setting.
2. P2/L82-83: How did you prepare the samples for RNASeq analysis? Which methods were followed? Please, provide more details even though the data was previously generated in another study.
3. P2/L85: Specify the “2 μg/mL”. I assume that these 2μg/mL are of LPS, right?
4. P2/L85: Define “DEGs”. It is the first time that appears in the text.
5. P2/L92: Be coherent with abbreviations. Abbreviations should be defined the first time they appear in the text (in this case, line 85).
Results:
1. P3/L116-117: I probably missed something, but there are DEGs between Control 0h and Control 24 h (points in red in Figure 1 between control 0h vs 24h), right? Why did you state that there are no significant differences between controls (0h vs 24h) if there are points with a p-value <0.05?
2. P3/L125: Data is duplicated in Figure 1. I would recommend representing 6 comparisons instead of 12 (6 comparisons twice) to clarify the figure. Include the description of the groups and the meaning of the black and red dots in the figure caption. Which is the p-value and fold change threshold used in the volcano plots?
3. P4/L126: I suggest including a new subsection “3.2” about the results of the comparison between Control 24 h and LPS-2.
4. P4/126: Which is the reason why you further analyzed Control 24 h vs LPS-2 and you did not include Control 24 h vs LPS-8 analysis?
5. P4/L134: According to the table caption, it seems that Table 1 includes all the DEGs found in the study but are only included the 752 genes involved in cell proliferation. Please clarify the table caption accordingly.
6. P4/L134: Capitalize the column “number” following the same criteria as the other columns of Table 1.
7. P4/L148-149: Specify in the Table 2 caption the comparison you used to extract the list of 12 genes.
8. P5/L157-158: Information about the function of the genes should be specified and discussed in the Discussion section, not in the Results section.
9. P5/L165: Table 3 concerns: Consider rephrasing the Table 3 caption as it does not include the common genes but the functions of the common genes.
I recommend removing the “DEGs” column in Table 3 because it does not provide additional information to the table (same 2035 DEGs in all functions).
I suggest writing the items of the “Functions” column in Table 3 in upper case letters to be more coherent with the rest of the items included (some are capitalized, and others aren’t).
Why is “VEGF” included in Table 3 as a “Functions” if VEGF is a gene?
10. P6/L167-169: I get confused with the classification used in Table 4, is it based on function or phenotype? Please be coherent in the text and the table caption.
11. P7/L176: “transcription factors” previously defined in line 100.
12. P7/L179: Why is stated that there are 8 different classes in Figure 2 and are depicted only 7?
13. P7/L180: If an abbreviation substitutes a plural name, include a suffix “s”. In this case, you are referring to “(…) most of the TF identified (…)” (plural). Please, revise other abbreviations accordingly through the text.
14. P7/L181-185: Information and interpretation of the results should be presented in the Discussion section, not in the Results section.
15. P7/187: Please, be coherent using abbreviations throughout the text. You previously defined “Transcription factors”.
16. P7/L198-200: Why did you include in the figure caption the number of TFs (1-10, 10-20, >20) in TF classes? I don’t get the point if you include the % in the figure.
17. P7/L202: Be coherent with the abbreviations (transcription factors). Please, revise this and other abbreviations for the entire manuscript. I will no longer mention it in the revision.
18. P8/L210-211: Interpretation of the results should be presented in the Discussion section, not in the Results section.
19. P8/L218-221: Long sentence, revise English grammar.
20. P8/L223: Why is Figure 4 presented before Figure 3?
21. P8/L223-224: Technically speaking, you did not create a pathway. The pathway was already “created” millions of years ago through evolution. You depicted a cell pathway and highlighted the DEGs based on your results.
22. P9/L228-229: Which is the meaning of the orange and blue lines in Figure 3? And the purple circles of “Bax” and “MDM2”?
23. P9/L332: I would recommend rephrasing the title of Figure 4. Which is the meaning of the red font in the figure (e.g., K-Ras, β-catenin, p53)?
Discussion:
1. P9/L236-237: Please, provide references for the statement “Previous studies have shown (…) in several tissues”.
2. P9/L237-238: Please, provide references for the statement “Numerous studies have revealed (…) inflammatory responses”.
3. P10/L241: Please, provide references for the statement “(…), which is consistent fully with previous studies”.
4. P10/L241-243: The sentence “So far, several intercellular (…) mechanism and pathways deeply.” is too general and vague to justify your research.
5. P10/L243-245: I recommend including more information about the present study in this sentence. In which type of cells was evaluated the change in gene expression? What was the inductor of such changes? In which species?
6. P10/L264: Use italics in the species name.
7. P10/L274-275: “increased” instead of “did increase”
8. P11/L296: The reference “(Koo et al. 2020)” does not follow the correct format according to the journal guidelines.
9. P11/L297: The reference “(Miao et al. 2020)” does not follow the correct format according to the journal guidelines.
Conclusions:
1. P11/L308-315: The conclusions should be rephrased according to the results. The conclusions are not supported by the results, and the conclusions are too speculative based on suggested mechanisms responsible for the observed transcriptomic changes. The authors did not test these hypotheses (because this was not the aim of the study). Which are the key findings of the study? Which are the implications of these results?
References:
1. References 16 and 25 are the same. Please, merge them and revise the manuscript.
Supplementary material:
1. Not all the biological pathways are clearly seen in the figure. Some of them are interrupted. Which is the meaning of the colors and length of the bars? The blue numbers are the p-values? Please, clarify the figure caption.
Reviewer 2 Report
This paper describes RNA-sequencing data by LPS dose-dependent treatment in the bovine endometrial epithelial cells.
Several differentially expressed genes (DEG) were detected between control and LPS treated group. This is very important to understand the mechanisms in the early implantation period. I judged that the paper could be published in the genes, if the following points were appropriately revised.
1) In some parts of all sentence, “over-expressed and under-expressed” are not means properly. I suggest that change “both of words” to “up-regulated or down-regulated”. Please correct them.
2) On page 4, line 129, it says "352 genes were under-expressed".
Even though displaying as green color in the Figure 3, but it was not explained at the discussion part. Among the most down-regulated genes, 1-2 genes are good to be explained in the discussion part. Addition of explanation is required.
Round 2
Reviewer 1 Report
The authors addressed most of questions and they improved the quality of the revised manuscript. I have a few minor recommendations.
P2/L83: Define BSA.
P2/L86-87: You have previously defined “bEEC” in line 73.
P2/L92: “into” instead of “at”.
P3/L104: “DEGs” instead of “differentially expressed genes”.
P3/L107: You have previously defined “DEGs” in line 100.
P3/L120: Reference 28 appears before reference 27 in the text. Please, revise and refresh the reference list to establish the correct order of the references.
P3/L138: “was” instead of “is”.
P4/L142-145: The first version of the figure is suitable for publication. “Significant means” instead of “Significant mean”.
P7/L204: “TFs” instead of “transcription factors” (same for line 217).
P8/L224-226: The previous version of these sentences was clearer.
P10/L288: According the cited reference 17, the units expressed are incorrect. Units should be “μg/mL” instead of “mg/ml”.
